# Impact of the Rapid Normalization of Chronic Hyperglycemia on the Receptor Activator of Nuclear Factor-Kappa B Ligand and the Osteoprotegerin System in Patients Living with Type 2 Diabetes: RANKL-GLYC Study

**DOI:** 10.3390/medicina58040555

**Published:** 2022-04-18

**Authors:** Dured Dardari, Claire Thomas, Francois-Xavier Laborne, Caroline Tourte, Elodie Henry, Megane Erblang, Stéphanie Bourdon, Alfred Penfornis, Philippe Lopes

**Affiliations:** 1Centre Hopitalier Sud Francilien, Department of Diabetes, Hôpital Sud Francilien, 9110 Corbeil-Essonnes, France; alfred.penfornis@chsf.fr; 2Laboratoire de Biologie de l’Exercice pour la Performance et la Santé (LBEPS), Univ Evry, Université Paris Saclay, 91025 Evry, France; claire.thomas@univ-evry.fr (C.T.); philippe.lopes@univ-evry.fr (P.L.); 3Centre Hospitalier Sud Francilien, Unité de Recherhce Clinique, Hôpital Sud Francilien, 9110 Corbeil-Essonnes, France; francois-xavier.laborne@chsf.fr (F.-X.L.); caroline.tourte@chsf.fr (C.T.); elodie.henry@chsf.fr (E.H.); 4Unité de Physiologie des Exercices et Activités en Conditions Extrêmes, Département Environnements Opérationnels, Institut de Recherche Biomédicale des Armées (IRBA), Université Paris Saclay, 91000 Evry, France; megane.erblang@univ-evry.fr; 5Paris-Sud Medical School, Université Paris-Saclay, 94270 Le Kremlin-Bicêtre, France; stephanie.bourdon@intradef.gouv.fr

**Keywords:** diabetes, Charcot neuroarthropathy, receptor activator of nuclear factor-kappa B ligand (RANKL), osteoprotegerin (OPG), HbA1c

## Abstract

The RANKL-GLYC study aims to explore the impact of the rapid correction of chronic hyperglycemia on the receptor activator of nuclear factor-kappa B ligand (RANKL) and its antagonist osteoprotegerin (OPG). RANKL and OPG are considered the main factors in the pathophysiology of Charcot neuroarthropathy, a devastating complication of the joints that remains poorly understood. The study began recruiting patients in September 2021 and ends in June 2022; the final study results are scheduled for January 2023.

## 1. Introduction

One of the major causes of diabetes and its associated complications is non-enzymatic glycation reaction [1,2,3]. Numerous studies conducted in recent years have highlighted the multiple pathophysiological features of diabetes complications [4,5]. It was previously observed in humans that the rapid normalization of hyperglycemia can cause neuropathy known as treatment-induced neuropathy in diabetes (TIND), which affects the small peripheral fibers [6]. This neuropathy manifests by characteristic pains (electric shocks, burning sensations) and disorders linked to the involvement of sympathetic and parasympathetic fibers (orthostatic hypotension, tachycardia, motor diarrhea). The pathophysiology of this phenomenon is not clear, although one study observed the presence of hypervascularization and inflammation in contact with the nerve endings, with the opening of arteriovenous shunts contemporary with symptoms [7].

People living with diabetes can develop a rare and devastating complication of the joints, known as Charcot neuroarthropathy (CN) [8], in circumstances similar to the onset of TIND following the rapid correction of hyperglycemia, although this theory remains to be confirmed [9,10]. The pathophysiology of CN is not yet completely understood, although it is associated with an activation of inflammation processes and bone remodeling markers [11], the disruption of the osteoblast and osteoclast system [12], the activation of the receptor activator of nuclear factor-kappa B ligand (RANKL) system and its antagonist osteoprotegerin (OPG) [13,14], and often stress fractures due to physical activity [15]. Inflammation and peripheral hypervascularization seem to be the common link between these two pathologies (TIND and CN).

## 2. Material and Methods

### 2.1. Study Objectives

The main study objective is to observe the RANKL levels of patients with balanced and unbalanced type 2 diabetes and analyze its evolution over 3 months, which corresponds to the period required for the correction of blood sugar levels in the unbalanced group.

The secondary research objectives of this study are to assess the evolution of autonomic neuropathy at inclusion (M0) and 3 months after the correction of HbA1c levels (M3) using a SUDOSCAN test [10] as well as to evaluate osteoprotegerin levels at M0 and M3.

### 2.2. Participants

All patients met the following inclusion criteria: male or female patients aged 18 to 70 years; type 2 diabetes for at least 1 year; and treatment with metformin, sulfonylurea, dipeptidyl peptidase-4 inhibitor, glucagon-like peptide 1 analogue, and/or insulin therapy with a stable antidiabetic regimen for at least 6 months. Premenopausal women used effective contraception and were monitored throughout the trial. All patients gave signed informed consent to participate in the study.

Patients in the unbalanced group (G1) were insufficiently balanced under this treatment regimen with HbA1c levels > 8.5% for at least 6 months.

Patients in the balanced group (G2) were balanced under this treatment with HbA1c levels < 7% for at least 6 months. Patients were matched with group G1 according to age (±5 years), sex, duration of diabetes (±2 years), and body mass index (BMI) (±2 kg/m^2^).

The exclusion criteria were as follows: insulin or antidiabetic therapy modified in the past 6 months; confirmed or suspected pregnancy; severe obesity (BMI > 35 kg/m^2^); other pathologies likely to interfere with blood sugar levels (e.g., corticosteroid use); existing CN or symptomatic autonomic neuropathy such as orthostatic hypotension or gastroparesis; preproliferative diabetic retinopathy; or anti-RANKL treatment.

### 2.3. Methods

The study was approved by the ethics committee of Nimes Hospital, France, on 30 June 2021 with approval number 2021.05.01 bis_ 21.03.15.39441.

### 2.4. Inclusion Visit (M0)

The investigator will propose the study to patients at a diabetology consultation, explain the study process, and provide them with the information note and consent form. After agreeing to participate in the study and providing signed consent, the inclusion visit will continue as follows: Pregnancy test: For women of childbearing age, a urine pregnancy test will be performed.Clinical examination: This examination will be carried out as part of usual care. Weight and height will be accurately recorded to determine BMI.Biological sampling: During the blood sample taken as part of treatment, an additional dry tube of 2 mL (EDTA tube) will be taken for this study. The dosage of glycated hemoglobin (HbA1c), C reactive protein, complete blood count, and creatinemia will be analyzed for treatment purposes, while RANKL and OPG will also be analyzed at M0.

SUDOSCAN^®^: The SUDOSCAN^®^ (Impeto Medical, Issy-les-Moulineaux, France) non-invasively measures the ability of sweat glands to release chloride ions in response to electrochemical stimulus with four electrodes placed on the palms of the hands and soles of the feet, as these areas have the highest density of sweat glands. The results of the small fiber conduction measurement are provided on a scale of 0–100 and considered normal if the conduction speed is >70. 

### 2.5. Research End Visit (M3)

During this visit at M3, the investigator will perform a clinical examination by measuring the weight and blood pressure of participants and taking a blood sample identical to that of the inclusion visit (M0). The SUDOSCAN test will again be performed for patients in both groups.

### 2.6. Statistical Methods

To our knowledge, no study to date has evaluated the RANKL levels in a population of patients with unbalanced type 2 diabetes before and after the correction of HbA1C levels. Therefore, we do not have previous data to make a precise assumption about the evolution and expected distribution of RANKL levels and perform a workforce calculation.

The size required for a randomized pilot study is controversial, with different methods being proposed [16]. As this is an observational study, to estimate the primary endpoint and perform parametric statistical tests, we need to include 30 patients per group (30 in the G1 group with high HbA1c and 30 in the comparative G2 group with normal HbA1c levels).

For each quantitative parameter, the descriptive analysis will include the mean, standard deviation, median, interquartile ranges, minimum, and maximum. Quantitative variables will be compared according to their distribution using the t-test or Wilcoxon test. Qualitative parameters will be expressed as numbers and their corresponding percentages. They will be compared using the Fisher’s exact test or chi2 test.

The initial RANKL levels of the G1 and G2 groups will be compared using a Student test. The variation in RANKL levels between M0 and M3 will be analyzed using a mixed linear model adjusted to the initial HbA1C levels, physical activity, and patient characteristics (sex, age, BMI, and diabetes duration).

All tests will be carried out bilaterally with a risk of first species alpha set at 5%. The analysis will be performed using R software (R version 4.0.2 (22 June 2020) © 2022, R Foundation for Statistical Computing Vienna, Austria).

The results will be stratified according to the reduction in HAb1c levels. The findings will first be presented globally and then according to two groups: (i) the group with a significant reduction in Hab1c levels but <2.5 points, and (ii) the group with a significant decrease in HBA1c levels of >2.5%.

## 3. Discussion

In a retrospective study that we recently published [17], we showed a significant drop in HbA1c levels in the 6 months prior to the onset of active CN. Nevertheless, in our patients, the rapid correction of HbA1c levels was not always accompanied by CN. While it is known that the rapid correction of chronic hyperglycemia can trigger TIND (though not systematically), the influence of this rapid correction on the onset of CN and the central nervous system has not yet been explored using SUDOSCAN [15]. In our study, a significant drop in HbA1c is expected for patients as part of their diabetes monitoring. Together with the main study objective, we will assess the impact of this reduction on the central nervous system via non-invasive tests. It is important to observe that this type of assessment was not performed in a previous study on TIND [11].

The pathogenic mechanisms of CN have long been the subject of debate. There are numerous competing theories, which we will summarize below. First, Mitchell and Charcot favored the so-called “neurovascular” theory, which suggests that increased blood flow to bone due to damaged “trophic nerves” leads to bone resorption and weakening, ultimately leading to fractures and deformities [18]. The regulation of blood flow and vasomotion in the skin of the lower limbs is preserved in the context of CN [19]. Elevated venous pressure in the foot was observed in both groups (17 participants with CN) compared to control subjects [20]. Clinical findings of a hot foot with dilated veins point to the presence of an arteriovenous shunt in CN [20]. However, using measurements obtained during a Doppler examination, other studies show no difference in microcirculation between a group with CN and a group of participants with neuropathy [21].

Second, Volkman and Virchow proposed the “neurotraumatic theory,” which suggests that the joints affected by CN undergo repeated trauma, resulting in complicated fractures and inducing deformity during healing. Eloesser [22] carried out experiments on cats. The dorsal roots of the spinal cord of 42 cats were ligated on one side; during 3 years of follow-up, the majority developed CN. Eloesser also subjected three cats to iatrogenic joint damage, leading to their development of typical CN within 3 weeks. As the physical properties of the bones, including “breaking strength,” did not change, he concluded that trauma played an important role in the genesis of CN [22]. 

Third, regarding the neuroinflammatory theory, in a review, Childs showed the existence of an association between diabetes mellitus and osteoporosis, which could contribute to the development of CN [23]. Patients with CN had reduced bone density in the lower limbs compared to neuropathic subjects [24]. Studies using bone markers to assess bone formation and resorption have demonstrated an increase in osteoclastic activity relative to osteoblastic activity in both acute and chronic forms of CN [25,26]. In 2007, Jeffcoate described CN as an increased inflammatory response to injury that induced increased bone lysis [27].

Taking the above elements into account, we may legitimately postulate that the rapid and significant correction of HbA1c levels may accompany the onset of active CN. The study of Xiang et al. demonstrated that the RANKL antagonist, OPG, is inhibited by the correction of hyperglycemia [28], which may explain the elevated RANKL levels observed during the active phase of CN [27].

## 4. Conclusions

The expected objectives of the RANKL/GLYC study may help to partially explain the onset of inflammation, which is described in the neuroinflammation theory as the spark inducing the acute phase of CN.

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
