# Peer review of "Impact of the Rapid Normalization of Chronic Hyperglycemia on the Receptor Activator of Nuclear Factor-Kappa B Ligand and the Osteoprotegerin System in Patients Living with Type 2 Diabetes: RANKL-GLYC Study"

_medicina, 2022, doi:10.3390/medicina58040555_

Round 1
Reviewer 1 Report
Major concern: Another control group should be added to the study. This group will include unbalanced diabetic patients who don't undergo rapid normalization of hyperglycemia as CN could be a delayed consequence of chronic uncontrolled hyperglycemia and not due to rapid normalization of hyperglycemia.
Minor concerns:
Few French words should be translated to English in lines 66 and 67 (Juin and apouval)
In line 114, a mistake in the abbreviation of Charcot neuroarthropathy; it shows as NC rather than CN.
Author Response
Dear Reviewer: First of all many thanks for all your comments, here are the answers we can give to your remarks
I
1- Another control group should be added to the study. This group will include unbalanced diabetic patients who don't undergo rapid normalization of hyperglycemia as CN could be a delayed consequence of chronic uncontrolled hyperglycemia and not due to rapid normalization of hyperglycemia.
Answer Dardari: Many thanks for your very interesting comment, we made an initial request to the ethics committee for the constitution of this group but our request was refused because the committee refuses to keep unbalanced patients for a period of 3 months for research reasons
2- Few French words should be translated to English in lines 66 and 67 (Juin and apouval) In line 114, a mistake in the abbreviation of Charcot neuroarthropathy; it shows as NC rather than CN.
Answer Dardari : We are very sorry for these errors, everything has been changed
The Manuscript entitled "Impact of the rapid normalization of chronic hyperglycemia on 2
the receptor activator of nuclear factor-kappa B ligand and oste- 3
oprotegerin system in patients living with type 2 diabetes: RANKL-GLYC study" discusses about the f chronic hyperglycemia on receptor activator of nuclear factor-kappa B ligand 13(RANKL) and its antagonist osteoprogesterin (OPG). RANKL and OPG are considered the main 14
factor in the pathophysiology of Charcot neuroarthropathy, a devastating complication of the joints 15
that remains poorly understood. The study starts recruiting patients in September 2021 and ends 16
in Jun 2022; final study results are scheduled for January 2023. this all is discussed in the Abstract section of the manuscript. However, this reviewer has serious concerns about the english grammar and language used in the present manuscript.
Answer Dardari Thank you for your comment , the manuscript set has been changed
Secondly This Manuscript is of only 2739 words and after checking the similarity from turnitin it was recorded to be 40% which is very high for such a short MS.
This paper needs overall improvement. I hope authors will improve all the things as per comment and then can resubmit. For the help to the authors, this reviewer has enclosed the plagiarism report performed from the turnit in software.
Answer Dardari Thank you for your comment , the manuscript set has been changed
Reviewer 2 Report
The Manuscript entitled "Impact of the rapid normalization of chronic hyperglycemia on 2
the receptor activator of nuclear factor-kappa B ligand and oste- 3
oprotegerin system in patients living with type 2 diabetes: 4
RANKL-GLYC study" discusses about the f chronic hyperglycemia on receptor activator of nuclear factor-kappa B ligand 13
(RANKL) and its antagonist osteoprogesterin (OPG). RANKL and OPG are considered the main 14
factor in the pathophysiology of Charcot neuroarthropathy, a devastating complication of the joints 15
that remains poorly understood. The study starts recruiting patients in September 2021 and ends 16
in Jun 2022; final study results are scheduled for January 2023.
this all is discussed in the Abstract section of the manuscript. However, this reviewer has serious concerns about the english grammar and language used in the present manuscript.
Secondly This Manuscript is of only 2739 words and after checking the similarity from turnitin it was recorded to be 40% which is very high for such a short MS.
This paper needs overall improvement. I hope authors will improve all the things as per comment and then can resubmit. For the help to the authors, this reviewer has enclosed the plagiarism report performed from the turnit in software.

Author Response
Dear Reviewer First of all many thanks for all your comments, here are the answers we can give to your remarks
Materials and methods state that RANKL levels are measured throughout the 3 month period but osteoprotegerin are only measured at M0 and M3. Why isn’t osteoprotegerin measured throughout the 3 month period?
Thank you for your comment: we have made the text clearer because RANKL and OPG will only be assayed at T0 and T3MONTH
Effective contraception” should be defined, as certain contraceptives alter hormone levels that impact bone function/structure. It would be interesting to include analysis on any potential effects of contraceptive method on outcome measures in the study. Like menopausal vs premenopausal status in women may affect outcome measures and should be considered in the study analysis.
Answer Dardari Thank you for your very interesting comment, we are going to do a subgroup study analysis for the group of women with contraception
Reviewer 3 Report
This study protocol describes the rationale, methods and possible implications for studying the neuropathic effects of hyperglycemia correction in diabetic patients. The proposed study is interesting and novel. It will add important information to the field of diabetes research.
Major comments:
- Materials and methods state that RANKL levels are measured throughout the 3 month period but osteoprotegerin are only measured at M0 and M3. Why isn’t osteoprotegerin measured throughout the 3 month period?
- “Effective contraception” should be defined, as certain contraceptives alter hormone levels that impact bone function/structure. It would be interesting to include analysis on any potential effects of contraceptive method on outcome measures in the study. Like menopausal vs premenopausal status in women may affect outcome measures and should be considered in the study analysis.
- Duration in which the diabetes has been uncontrolled in G2 and the “degree” (HbA1c and hyperglycemic levels) of the diabetes may affect the presence of potential symptoms, such as neuropathy. Thus, including patients with HbA1c > 8.5% for at 6 least months may create large amounts of variation in the data. If unable to define these criteria more concretely by using inclusion ranges, authors should consider stratifying data based on HbA1c and duration for diabetes has not been controlled in data analysis to determine if different outcomes may be present based on these factors.
- “Clinical examination” should be defined. What is performed as part of the examination and who performs the exam? Does the same physician/investigator perform each exam? If not, how were they standardized across patients?
- The statistical methods sections states that RANKL analyses will be adjusted by physical activity levels, however the methods section does not include the methodology for determining physical activity levels in patients.
- It is unclear why DXA scans were not included to assess bone mineral density. This should be added into the discussion.
Minor comments:
- There are many grammatical and formatting errors. The manuscript could benefit from editing by a native English speaker.
- “Rapid normalization” should be clarified as a specific time period.
- SUDOSCAN, SU, DPP4 and GLP1 should be defined at first use.
- Change “women of childbearing age” to premenopausal women.
- Is blood collected in EDTA or heparin coated tubes?
Author Response
Dear Reviewer First of all many thanks for all your comments, here are the answers we can give to your remarks
Duration in which the diabetes has been uncontrolled in G2 and the “degree” (HbA1c and hyperglycemic levels) of the diabetes may affect the presence of potential symptoms, such as neuropathy. Thus, including patients with HbA1c > 8.5% for at 6 least months may create large amounts of variation in the data. If unable to define these criteria more concretely by using inclusion ranges, authors should consider stratifying data based on HbA1c and duration for diabetes has not been controlled in data analysis to determine if different outcomes may be present based on these factors.
Answer Dardari Thank you once again for your comment, the startification of the results according to the acquired HbA1c level is thanks to your comment
Clinical examination” should be defined. What is performed as part of the examination and who performs the exam? Does the same physician/investigator perform each exam? If not, how were they standardized across patients?
Answer Dardari The same practitioner will examine the patients at the inclusion visit and at the end of the study, the details of the clinical examination have been added
The statistical methods sections states that RANKL analyses will be adjusted by physical activity levels, however the methods section does not include the methodology for determining physical activity levels in patients.
Answer Dardari The paragraph has been changed and the physical activity has been removed sorry for this error
It is unclear why DXA scans were not included to assess bone mineral density. This should be added into the discussion.
Answer Dardari Thank you for your comment, we think that there is little chance that the DXA will be sensitive to the modifications in our study over a period of 3 months
There are many grammatical and formatting errors. The manuscript could benefit from editing by a native English speaker.“Rapid normalization” should be clarified as a specific time period. SUDOSCAN, SU, DPP4 and GLP1 should be defined at first use. Change “women of childbearing age” to premenopausal women
Answer Dardari all modification were done, were are really sorry
Is blood collected in EDTA or heparin coated tubes?
Answer Dardari We use ETDA tubes
Round 2
Reviewer 1 Report
In Line 21: It is June not Jun 2022
Author Response
Many thanks for your comment
modification done
Kindly regards
Dured Dardari MD PhD
Reviewer 2 Report
The MS entitled "Impact of the rapid normalization of chronic hyperglycemia on 2
the receptor activator of nuclear factor-kappa B ligand and oste- 3
oprotegerin system in patients living with type 2 diabetes: 4
RANKL-GLYC study" may be considered following the revision.
- overall the language of the paper and the grammar should be improved throughout the manuscript.
- There are instances in introduction wherein authors didn't provided the complete information about diabetes and their references.
Introduce these sentences in the introduction:
one of the major reason for diabetes and its associated complications is non-enzymatic glycation reaction [1-3]. There are number of studies conducted in recent before which holds true for multiple pathophysiological condition [4-5].
[1]. A Glycation Angle to Look into the Diabetic Vasculopathy: Cause and Cure. Current vascular pharmacology. 2017;15(4):352-64.
[2]. An immunohistochemical analysis to validate the rationale behind the enhanced immunogenicity of D-ribosylated low density lipo-protein. PLoS One, 9(11), p.e113144.
[3]. Autoimmune response to AGE modified human DNA: Implications in type 1 diabetes mellitus. Journal of Clinical & Translational Endocrinology. 2014 Jun 25;1(3):66-72.
[4]. Prevalence of auto-antibodies against D-ribose-glycated-hemoglobin in diabetes mellitus. Glycobiology. 2019 May 1;29(5):409-18.
[5]. A biochemical & biophysical study on in-vitro anti-glycating potential of iridin against d-Ribose modified BSA.
Author Response
Many thanks for your commentes
We added all your observations in our new manuscript version
Kindly regards
Dured Dardari